# Skipping Breakfast and Incidence of Frequent Alcohol Drinking in University Students in Japan: A Retrospective Cohort Study

**DOI:** 10.3390/nu14132657

**Published:** 2022-06-27

**Authors:** Yuichiro Matsumura, Ryohei Yamamoto, Maki Shinzawa, Taisuke Matsushita, Ryuichi Yoshimura, Naoko Otsuki, Masayuki Mizui, Isao Matsui, Junya Kaimori, Yusuke Sakaguchi, Chisaki Ishibashi, Seiko Ide, Kaori Nakanishi, Makoto Nishida, Takashi Kudo, Keiko Yamauchi-Takihara, Izumi Nagatomo, Toshiki Moriyama

**Affiliations:** 1Health Promotion and Regulation, Department of Health Promotion Medicine, Graduate School of Medicine, Osaka University, Toyonaka 560-0043, Japan; matsumura@hacc.osaka-u.ac.jp (Y.M.); moriyama@wellness.hss.osaka-u.ac.jp (T.M.); 2Department of Nephrology, Graduate School of Medicine, Osaka University, Suita 565-0871, Japan; shinzawa@kid.med.osaka-u.ac.jp (M.S.); t.matsushita@kid.med.osaka-u.ac.jp (T.M.); mmizui@kid.med.osaka-u.ac.jp (M.M.); matsui@kid.med.osaka-u.ac.jp (I.M.); 3Health and Counseling Center, Osaka University, Toyonaka 560-0043, Japan; r84ysmr@med.shimane-u.ac.jp (R.Y.); otsuki@hacc.osaka-u.ac.jp (N.O.); cishibashi@hacc.osaka-u.ac.jp (C.I.); ide@cardiology.med.osaka-u.ac.jp (S.I.); k-nakanishi@hacc.osaka-u.ac.jp (K.N.); mnishida@wellness.hss.osaka-u.ac.jp (M.N.); kudo@psy.med.osaka-u.ac.jp (T.K.); takihara@wellness.hss.osaka-u.ac.jp (K.Y.-T.); iznagatomo@hacc.osaka-u.ac.jp (I.N.); 4Division of Nephrology, Shimane University Hospital, Izumo 693-8501, Japan; 5Department of Inter-Organ Communication Research in Kidney Diseases, Graduate School of Medicine, Osaka University, Suita 565-0871, Japan; kaimori@kid.med.osaka-u.ac.jp (J.K.); sakaguchi@kid.med.osaka-u.ac.jp (Y.S.)

**Keywords:** university students, skipping breakfast, frequency of alcohol drinking

## Abstract

Frequency of alcohol drinking is a potential predictor of binge drinking of alcohol, a serious social problem for university students. Although previous studies have identified skipping breakfast as a predictor of various health-compromising behaviors and cardiometabolic diseases, few studies have assessed the association between skipping breakfast and the incidence of frequent alcohol drinking. This retrospective cohort study included 17,380 male and 8799 female university students aged 18–22 years admitted to Osaka universities between 2004 and 2015. The association between breakfast frequency (eating every day, skipping occasionally, and skipping often/usually) and the incidence of frequent alcohol drinking, defined as drinking ≥4 days/week, was assessed using multivariable-adjusted Cox proportional hazards models. During the median observational period of 3.0 years, 878 (5.1%) men and 190 (2.2%) women engaged in frequent alcohol drinking. Skipping breakfast was significantly associated with the incidence of frequent alcohol drinking (adjusted hazard ratios [95% confidence interval] of eating every day, skipping occasionally, and skipping often/usually: 1.00 [reference], 1.02 [0.84–1.25], and 1.48 [1.17–1.88] in men; 1.00 [reference], 1.60 [1.03–2.49], and 3.14 [1.88–5.24] in women, respectively). University students who skipped breakfast were at a higher risk of frequent alcohol drinking than those who ate breakfast every day.

## 1. Introduction

Excessive alcohol consumption is a global health and social burden because it is associated with suicide [1], death by injury [2], liver cirrhosis [3], ischemic heart disease [4], and cancers, including those of the oral cavity and pharynx, esophagus, colorectum, liver, larynx, and female breast [5]. Binge drinking, a type of excessive alcohol consumption defined as consuming five or more standard drinks per occasion for men and four or more drinks for women [6], is common in university students in many countries; 43–44%, 56%, 76%, 26–28%, 23%, and 53% of university students had a recent history of binge drinking in the US [7], the UK [8], Switzerland [9], China [10], and Japan [11], respectively. Recently, binge drinking in university students has been regarded as a serious social problem in these countries [11,12], because multiple studies showed that university students with a binge drinking habit were at high risk of missed classes and lower grades [13]; injuries and sexual assaults [14]; non-medical use of prescription drugs [15]; memory blackouts [16]; changes in brain function [17]; lingering cognitive deficits [18]; and even death [14]. One of the key predictors of binge drinking in young adults is the frequency of alcohol drinking. A large cross-sectional study, including 10,466 current drinkers aged 18–76 years in Canada, reported that drinkers drinking alcohol ≥3 days/week were more vulnerable to binge drinking than those drinking alcohol <3 days/week and, additionally, suggested that the association between frequency of alcohol drinking and binge drinking was stronger in younger drinkers than that in older drinkers [19]. Frequency of alcohol drinking may be a potential target for suppressing the deleterious effects of binge drinking on young adults.

Breakfast is the most important meal of the day. Japanese people mainly eat rice or bread at breakfast [20]. Multiple studies have identified skipping breakfast as a risk factor for cardiometabolic diseases, including obesity [21], high blood pressure [22], high low-density lipoprotein cholesterol level [23], type 2 diabetes [24], proteinuria [25], and cardiovascular disease [26]. Skipping breakfast is common in university students [27]. The prevalence of skipping breakfast in university students was 32%, 9–16%, 33%, 46–69%, and 17–31% in the US [28], Spain [29,30], Australia [31], China [32,33], and Japan [34,35], respectively. Some studies reported that skipping breakfast was associated with health-compromising behaviors, such as smoking or alcohol drinking [36], in addition to affecting test grades and school attendance [37]. Although several cross-sectional studies have suggested an association between skipping breakfast and alcohol drinking in university students, few cohort studies have confirmed this association [38,39].

The present 5-year retrospective cohort study aimed to clarify the association between skipping breakfast and the incidence of frequent alcohol drinking in 26,179 university students at a single national university in Japan, where drinking alcohol under the age of 20 is prohibited by law. The findings of the present study clarify the clinical impact of skipping breakfast on the frequency of alcohol drinking in young adults.

## 2. Materials and Methods

### 2.1. Participants

The eligible participants of the present study were 38,111 university students admitted to Osaka University, one of the largest national universities in Japan, between 2004 and 2015. The participants underwent baseline health checkups on admission in April or October between 2004 and 2015. We excluded 229 (0.6%) students aged <18 and >22 years, 36 (0.1%) who drank ≥4 days/week at baseline, 8918 (23.4%) with missing baseline data, and 2749 (7.2%) who had no measurement of drinking status after their baseline checkups during the 5-year observation period. Finally, we included 26,179 (68.7%) students to assess the association between skipping breakfast and the incidence of frequent alcohol drinking (Figure 1).

The study protocol was approved by the ethics committee of the Health and Counseling Center of Osaka University (No. 5, 2022) and Osaka University Hospital (No. 18352–3). The present study used an opt-out approach to provide informed consent according to the Japanese Ethical Guidelines for Medical and Health Research Involving Human Subjects. All procedures in this study involving human participants were performed in accordance with the ethical standards of the institutional and national research committee and with the 1964 Helsinki Declaration and its later amendments or comparable ethical standards.

### 2.2. Measurements

All data were retrieved from the electronic database of the Health and Counseling Center of Osaka University. The baseline variables at the health checkup on admission included age, body mass index (BMI = body weight (kg)/height^2^ (m^2^)), and general questionnaires regarding breakfast frequency, frequency of drinking alcohol, and potential confounders, including smoking status [40,41], stress frequency [42,43], weekday sleep duration [44,45], and living arrangements [27,46]. Breakfast frequency was determined according to the question “How often did you have breakfast during the past year?”, for which the possible responses were “eating almost every day”, “skipping occasionally”, “skipping often”, and “skipping usually”. Frequency of alcohol drinking was based on the question, “How often did you drink the past year?’’ with five possible responses: “I did not drink”, “I drank occasionally”, “I drank a day/week”, “I drank 2–3 days/week”, and “I drank ≥4 days/week”. Drinkers were defined as those who drank alcohol occasionally or more. Frequent alcohol drinking was defined as drinking ≥4 days/week. Smoking status was determined according to the question “Do you smoke?”; the possible answers were “I do not smoke”, “I quit smoking”, “I would like to quit smoking”, and “I smoke”. Because most students answered “I do not smoke” (male and female students, *n* = 17,335 [99.7%] and 8790 [99.9%], respectively), smoking status was categorized into two categories: non-smokers with the answer of “I do not smoke”, and smokers with other answers. Weekday sleep duration was ascertained by the question “How long do you sleep on weekdays?” with five possible responses of <5, 5–6, 6–7, 7–8, and ≥8 h. Living arrangement was asked using the question “Which is your living arrangement?” with four possible answers: living with family, living alone, living in a dormitory, and other living arrangements. Stress frequency was determined by the question “Do you feel stressed?” with four possible answers: “I rarely feel stressed”, “I sometimes feel stressed”, “I often felt stressed”, and “I always feel stressed.”

The outcome measure of the present study was the incidence of frequent alcohol drinking, which was defined as ≥4 days/week of alcohol drinking based on the questionnaire at the annual health checkups during the 5-year observational period. We set the outcome at ≥4 days/week of alcohol drinking because the previous Canadian cohort study showed that adults with ≥3 day/week of alcohol drinking were vulnerable to binge drinking [19]. The observational period was defined as the period from the baseline health checkup on admission to (i) the incidence of the outcome or (ii) the last answer to the questionnaire on frequency of alcohol drinking at the annual health checkups within 5 years (= 1825 days) after the baseline checkup.

### 2.3. Statistical Analysis

Owing to the small number of students usually skipping breakfast (male and female students, *n* = 201 [1.2%] and 76 [0.9%], respectively), we combined those usually skipping breakfast and those skipping often into a single category. Thus, breakfast frequency was recategorized into three groups: eating every day, skipping occasionally, and skipping often/usually.

All analyses were performed separately for male and female students. Baseline characteristics of the three groups of breakfast frequency were compared using analysis of variance or the chi-square test, as appropriate. The cumulative probabilities of frequent alcohol drinking in the three breakfast frequency groups were calculated using the Kaplan–Meier method and compared using the log-rank test. To assess the association between breakfast frequency and the incidence of frequent alcohol drinking, unadjusted and adjusted hazard ratios (HRs) were calculated using unadjusted and multivariable-adjusted Cox proportional hazard models. Multivariable-adjusted Cox proportional hazards models included age (18, 19, 20, 21, and 22 years), BMI (kg/m^2^), smoking status (for male students), living arrangements (living with family, living alone, living in dormitories, and other living arrangements), weekday sleep duration (<5, 5–6, 6–7, 7–8, and ≥8 h), and stress frequency (rarely, sometimes, often, and always) as covariates. For female students, smoking status was not included in the multivariable-adjusted model because of the small number of smokers (*n* = 9). The proportional hazard assumption was tested using the Schoenfeld residuals.

Continuous variables are expressed as mean ± standard deviation or median (25–75%), as appropriate, and categorical variables are expressed as numbers with proportions. Statistical significance was set at *p* < 0.05. Statistical analyses were performed using Stata version 16.1 (Stata Corp, College Station, TX, USA).

## 3. Results

The baseline characteristics of 17,380 male students are presented in Table 1. Compared with 14,115 (81.2%) male students eating breakfast every day, 1030 (5.9%) male students skipping breakfast often/usually were likely to be older, have lower BMI levels, have a higher prevalence of smokers and drinkers, sleep longer on weekdays, and feel stressed more frequently (*p* < 0.05). Similar tendencies were observed in 8799 female students (Table 2).

During the median observational period of 3.03 (3.01–3.06) years, 686 (4.9%), 116 (5.2%), and 76 (7.4%) male students who ate breakfast every day, skipped it occasionally, and skipped it often/usually, respectively, engaged in frequent alcohol drinking (Table 3 and Figure 2a). An unadjusted model showed that male students skipping breakfast were at a higher risk of frequent alcohol drinking than those eating every day (unadjusted HR of eating every day, skipping occasionally, and skipping often/usually: 1.00 [reference], 1.06 [95% confidence interval, 0.87–1.30], and 1.52 [1.20–1.93], respectively) (Table 3). Moreover, even after adjusting for clinically relevant factors, skipping breakfast often/usually was significantly more associated with the incidence of frequent alcohol drinking than eating every day (adjusted HR of eating every day, skipping occasionally, and skipping often/usually: 1.00 [reference], 1.02 [0.84–1.25], and 1.48 [1.17–1.88], respectively) (Table 3).

Of the 8799 female students, the incidence of frequent alcohol drinking was observed in 150 (1.9%), 23 (3.0%), and 17 (6.1%) students eating breakfast every day, skipping it occasionally, and skipping it often/usually, respectively, during the median observational period of 3.04 years (3.01–3.06) (Table 3). Compared with students who ate breakfast every day, the cumulative probability of frequent alcohol drinking was significantly higher in students who skipped occasionally and often/usually (Figure 2b). In an unadjusted model, skipping breakfast occasionally and often/usually was significantly associated with the incidence of frequent alcohol drinking (unadjusted HR of eating every day, skipping occasionally, and skipping often/usually: 1.00 [reference], 1.63 [1.05–2.52], and 3.15 [1.91–5.20], respectively) (Table 3). A multivariable-adjusted model identified skipping occasionally and often/usually as significant predictors of frequent drinking (adjusted HR of eating almost every day, skipping occasionally, skipping often/usually: 1.00 [reference], 1.60 [1.03–2.49], and 3.14 [1.88–5.24], respectively) (Table 3).

## 4. Discussion

The present retrospective cohort study identified skipping breakfast as a significant predictor of frequent alcohol drinking in university students. Some advantages of the present study were the cohort study design with the 5-year observational period and the large sample size (*n* = 26,179), which enabled statistically meaningful analyses to assess the clinical impact of skipping breakfast on the incidence of frequent alcohol drinking in both male and female students.

Several large cross-sectional studies have reported the association between skipping breakfast and the prevalence of alcohol consumption. A Chinese cross-sectional study including 19,542 school-aged adolescents reported that female adolescents who skipped breakfast were more likely to drink during the past 7 days than those who did not skip breakfast, whereas no association between skipping breakfast and the prevalence of drinking was observed in male adolescents [39]. Another international cross-sectional study, including 21,972 university students in 28 countries, reported an association between the frequency of skipping breakfast and the prevalence of binge drinking [38]. These cross-sectional studies suggest an association between skipping breakfast and alcohol consumption. However, only a few cohort studies have ascertained the clinical impact of skipping breakfast on alcohol consumption. The present retrospective cohort study, which included 17,380 male and 8799 female university students, verified that skipping breakfast predicted the incidence of frequent alcohol drinking in both male and female students (Table 3 and Figure 2), suggesting that university students who skip breakfast might be vulnerable to the detrimental effects of excessive alcohol consumption.

One plausible mechanism for the association between skipping breakfast and frequency of alcohol drinking may be depression. Skipping breakfast is a clinical predictor of depression. A 1-year prospective Chinese cohort study, including 757 university students, reported that skipping breakfast, defined as breakfast consumption of ≤1, 2–5, vs. ≥6 times/week, was associated with the incidence of self-reported depressive symptoms based on the Zung Self-Rating Depression Scale in a dose-dependent manner [39]. A similar dose-dependent association between skipping breakfast and the incidence of depression was observed in 115,217 Chinese adolescents aged 12–18 years [47] and 716 Japanese adults aged 19–68 years [48]. These studies suggest that students who skipped breakfast were at a higher risk of depression in the present study. Young patients with depressive disorders may be more vulnerable to frequent drinking. A Finnish cohort study, including 1545 adolescent twins, showed that those diagnosed with depressive disorders, using the Diagnostic and Statistical Manual of Mental Disorders, fourth edition (DSM-IV), were at a higher risk of frequent use of alcohol [49]. Other cohort studies, including adolescents [50,51] and university students [52], suggested that depressive disorders predict excessive alcohol use. Therefore, given the results of these previous studies, university students who skipped breakfast might be vulnerable to depression and were at a higher risk of frequent alcohol drinking in the present study. Further studies are essential to assess the underlying mechanisms linking skipping breakfast with the incidence of frequent alcohol drinking.

The present study had several limitations. First, because alcohol consumption at each drinking occasion was not available in the present study, the clinical impact of skipping breakfast on the frequency of binge drinking was not assessed in the present study. An international cross-sectional study including 21,972 university students in 28 countries showed that skipping breakfast was associated with the prevalence of binge drinking [20]. However, few cohort studies have assessed the association between skipping breakfast and the incidence of binge drinking, which should be clarified in future studies. Second, the threshold of the frequency of skipping breakfast for the prediction of the incidence of frequent alcohol drinking was not clear in the present study because of the questionnaire of skipping breakfast with subjective responses of “Eating almost every day”, “Skipping occasionally”, “Skipping often”, and “Usually skipping.” Although the present study could not assess a cutoff value for the frequency (days/week) of skipping breakfast to predict the incidence of frequent alcohol drinking, it clarified the dose-dependent association between breakfast frequency and the incidence of frequent alcohol drinking. Our previous retrospective cohort study, using the same questionnaire for breakfast frequency, disclosed a similar dose-dependent association between breakfast frequency and the incidence of proteinuria in 5439 female workers [25]. The results of the present study and our previous study strongly suggest the subjective questionnaire for breakfast frequency have a clinical relevance. The optimal cutoff value of the frequency (day/week) of skipping breakfast should be assessed in future studies. Third, the present study included university students from a single national university in Japan. The generalizability of the findings of this study should be confirmed in different cohorts. Fourth, because of the nature of the observational study design of the present study, the association between breakfast frequency and the incidence of frequent alcohol drinking was critically confounded by unmeasured factors. The clinical impact of breakfast frequency should be verified by randomized controlled trials.

## 5. Conclusions

The present retrospective cohort study, which included 26,179 university students at a single university, identified skipping breakfast as a significant predictor of the incidence of frequent alcohol drinking. These findings provide clinically useful information to identify university students potentially vulnerable to excessive alcohol consumption. A well-designed study is essential to verify the clinical impact of skipping breakfast on alcohol use disorders in young adults.

## Figures and Tables

**Figure 1 nutrients-14-02657-f001:**
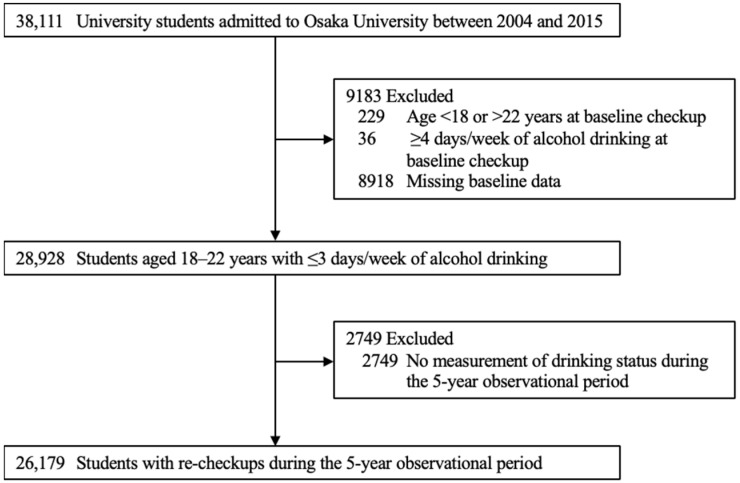
Flow diagram of inclusion and exclusion of study participants.

**Figure 2 nutrients-14-02657-f002:**
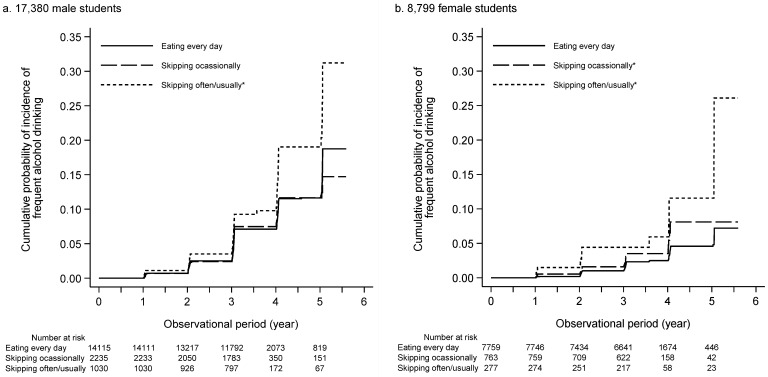
Baseline breakfast frequency and the incidence of frequent alcohol drinking in 17,380 male (**a**) and 8799 female (**b**) students. * *p* < 0.05 (vs. eating every day).

**Table 1 nutrients-14-02657-t001:** Baseline characteristics of 17,380 male university students stratified by breakfast frequency.

Breakfast Frequency	All	Eating Every Day	Skipping Occasionally	Skipping Often/Usually
Number	17,380	14,115	2235	1030
Age: 18 years, *n* (%)	11,280 (64.9)	9398 (66.6)	1312 (58.7)	570 (55.3)
19	5583 (32.1)	4386 (31.1)	808 (36.2)	389 (37.8)
20	378 (2.2)	259 (1.8)	80 (3.6)	39 (3.8)
21	99 (0.6)	51 (0.4)	24 (1.1)	24 (2.3)
22	40 (0.2)	21 (0.2)	11 (0.5)	8 (0.8)
Body mass index, kg/m^2^	21.6 ± 2.9	21.6 ± 2.9	21.5 ± 2.9	21.2 ± 2.9
Smokers, *n* (%)	45 (0.3)	20 (0.1)	18 (0.8)	7 (0.7)
Drinkers, *n* (%)	1578 (9.1)	1107 (7.8)	304 (13.6)	167 (16.2)
Sleep duration: <5 h, *n* (%)	536 (3.1)	417 (3.0)	68 (3.0)	51 (5.0)
5–6	5540 (31.9)	4519 (32.0)	698 (31.2)	323 (31.4)
6–7	8401 (48.3)	6910 (49.0)	1058 (47.3)	433 (42.0)
7–8	2487 (14.3)	1995 (14.1)	323 (14.5)	169 (16.4)
≥8	416 (2.4)	274 (1.9)	88 (3.9)	54 (5.2)
Living arrangement:				
Living with family, *n* (%)	8341 (48.0)	6939 (49.2)	931 (41.7)	471 (45.7)
Living alone	7726 (44.5)	6133 (43.5)	1112 (49.8)	481 (46.7)
Living in dormitory	1083 (6.2)	860 (6.1)	157 (7.0)	66 (6.4)
Other living arrangements	230 (1.3)	183 (1.3)	35 (1.6)	12 (1.2)
Stress frequency: rarely, *n* (%)	4499 (25.9)	3761 (26.6)	495 (22.1)	243 (23.6)
sometimes	10,110 (58.2)	8206 (58.1)	1341 (60.0)	563 (54.7)
often	2431 (14.0)	1899 (13.5)	347 (15.5)	185 (18.0)
always	340 (2.0)	249 (1.8)	52 (2.3)	39 (3.8)

Mean ± standard deviation. *p* < 0.05 for all variables.

**Table 2 nutrients-14-02657-t002:** Baseline characteristics of 8799 female university students stratified by breakfast frequency.

Breakfast Frequency	All	Eating Every Day	Skipping Occasionally	Skipping Often/Usually
Number	8799	7759	763	277
Age: 18 years, *n* (%)	6524 (74.1)	5822 (75.0)	534 (70.0)	168 (60.6)
19	2047 (23.3)	1785 (23.0)	187 (24.5)	75 (27.1)
20	157 (1.8)	111 (1.4)	25 (3.3)	21 (7.6)
21	47 (0.5)	26 (0.3)	11 (1.4)	10 (3.6)
22	24 (0.3)	15 (0.2)	6 (0.8)	3 (1.1)
Body mass index, kg/m^2^	20.5 ± 2.4	20.5 ± 2.4	20.3 ± 2.4	20.4 ± 2.4
Smokers, *n* (%)	9 (0.1)	6 (0.1)	2 (0.3)	1 (0.4)
Drinkers, *n* (%)	393 (4.5)	298 (3.8)	59 (7.7)	36 (13.0)
Sleep duration: <5 h, *n* (%)	267 (3.0)	214 (2.8)	35 (4.6)	18 (6.5)
5–6	3144 (35.7)	2779 (35.8)	273 (35.8)	92 (33.2)
6–7	4181 (47.5)	3734 (48.1)	336 (44.0)	111 (40.1)
7–8	1090 (12.4)	944 (12.2)	103 (13.5)	43 (15.5)
≥8	117 (1.3)	88 (1.1)	16 (2.1)	13 (4.7)
Living arrangement:				
Living with family, *n* (%)	4586 (52.1)	4103 (52.9)	349 (45.7)	134 (48.4)
Living alone	3396 (38.6)	2944 (37.9)	331 (43.4)	121 (43.7)
Living in dormitory	616 (7.0)	534 (6.9)	66 (8.7)	16 (5.8)
Other living arrangements	201 (2.3)	178 (2.3)	17 (2.2)	6 (2.2)
Stress frequency: rarely, *n* (%)	1800 (20.5)	1637 (21.1)	116 (15.2)	47 (17.0)
sometimes	5350 (60.8)	4714 (60.8)	476 (62.4)	160 (57.8)
often	1479 (16.8)	1274 (16.4)	144 (18.9)	61 (22.0)
always	170 (1.9)	134 (1.7)	27 (3.5)	9 (3.2)

Mean ± standard deviation. *p* < 0.05 for all variables.

**Table 3 nutrients-14-02657-t003:** Associations between breakfast frequency and the incidence of frequent drinking.

	Eating Every Day	Skipping Occasionally	Skipping Often/Usually
Male students			
Incidence of frequent drinking, *n* (%)	686 (4.9)	116 (5.2)	76 (7.4)
Observational period, year	3.03 (3.01–3.06)	3.03 (3.01–3.06)	3.03 (3.01–3.06)
IR per 1000 PY (95% CI)	16.1 (14.9–17.4)	17.4 (14.5–20.9)	25.0 (20.0–31.4)
Unadjusted HR (95% CI)	1.00 (Reference)	1.06 (0.87–1.30)	1.52 (1.20–1.93)
Adjusted HR (95% CI) *	1.00 (Reference)	1.02 (0.84–1.25)	1.48 (1.17–1.88)
Female students			
Incidence of frequent drinking, *n* (%)	150 (1.9)	23 (3.0)	17 (6.1)
Observational period, year	3.04 (3.02–3.06)	3.04 (3.01–3.06)	3.04 (3.01–3.06)
IR per 1000 PY (95% CI)	6.17 (5.3–7.2)	9.88 (6.6–14.9)	20.2 (12.6–32.5)
Unadjusted HR (95% CI)	1.00 (Reference)	1.63 (1.05–2.52)	3.15 (1.91–5.20)
Adjusted HR (95% CI) *	1.00 (Reference)	1.60 (1.03–2.49)	3.14 (1.88–5.24)

Median (25–75%). CI, confidence interval; IR, incidence rate; HR, hazard ratio; PY, person years. * Adjusted for age (18, 19, 20, 21, and 22 years), body mass index (kg/m^2^), smoking status (for male students), living arrangements (living with family, living alone, living in dormitories, and other living arrangements), weekday sleep duration (<5, 5–6, 6–7, 7–8, and ≥8 h), and stress (rarely, sometimes, often, and always).

## Data Availability

The data presented in this study are available upon request from the corresponding author. The data are not publicly available because they were not originally collected for research purposes.

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
