# Peer review of "Skipping Breakfast and Incidence of Frequent Alcohol Drinking in University Students in Japan: A Retrospective Cohort Study"

_nutrients, 2022, doi:10.3390/nu14132657_

Round 1
Reviewer 1 Report
Dear Authors,
Thank you for the opportunity to review this paper examining the skipping breakfast and incidence of frequent drinking in university students. To enhance the readability and wider context for the audience, I have some points below which I believe should be addressed to make the paper stronger.
Abstract:
Information on the age of the respondents is missing in the Abstract. It is also worth providing criteria for the frequency of alcohol consumption and skipping breakfast.
Introduction:
Here, it is useful to provide country-specific data on breakfast consumption and frequency of alcohol consumption - if available.
General remark:
On the basis of the conducted analysis, it is possible to estimate only the existence of a relationship between the frequency of alcohol consumption and the frequency of skipping breakfast. Unfortunately, the cause-and-effect relationship cannot be assessed. Because maybe it is the more frequent consumption of alcohol that causes skipping breakfasts? Which may be logical, because with symptoms of excessive alcoholism, reluctance to eat is typical.
Materials and methods:
I do not understand the reason for excluding students who consume alcohol ≥ 4 times a week?
It is not clear to me why questions were asked about sleep duration, stress levels or smoking? Have relationships between these factors been studied? If so, it should be noted in Introduction. If not, then perhaps this information should be omitted.
Line 123 (and in Table 1,2,3): “skipping ≥ often” – please explain?
Results:
There are different fonts size in tables 1, 2 and 3.
Due to the very small numbers, it may not be worth analyzing the data into age categories (e.g. in the case of female skipping breakfast ≥often only 3!). Maybe it is worth doing an analysis for the age groups, e.g. 18-19, 20-22?
The results of the statistical analysis should be shown in the tables!
Discussion:
For the reader's convenience, it would be better to distinguish the Conclusions as a separate paragraph.
Author Response
We appreciate many kind comments by the reviewer on our manuscript. All comments greatly help us improve the quality of our study. Now we hope that our revised manuscript was at the level of publication in Nutrients.
1. Information on the age of the respondents is missing in the Abstract. It is also worth providing criteria for the frequency of alcohol consumption and skipping breakfast.
We thank the reviewer a good advice. We added the age of the participants and define the breakfast frequency and the frequent drinking of alcohol as follows:
“(Line 23) This retrospective cohort study included 17,380 male and 8,799 female university students aged 18–22 years admitted to Osaka universities between 2004 and 2015. The association between breakfast frequency (eating every day, skipping occasionally, and skipping often/usually) and the incidence of frequent alcohol drinking, defined as drinking ≥4 days/week, was assessed using multivariable-adjusted Cox proportional hazards models.”
2. Here, it is useful to provide country-specific data on breakfast consumption and frequency of alcohol consumption - if available.
We appreciate the nice advice by the reviewer. We added the prevalence of binge drinking and skipping breakfast as follows.
"(Line 39) Binge drinking, a type of excessive alcohol consumption defined as consuming five or more standard drinks per occasion for men and four or more drinks for women [6], is common in university students in many countries: 43-44%, 56%, 76%, 26-28%, 23%, and 53% of university students had a recent history of binge drinking in the US [7], the UK [8], Switzerland [9], China [10], and Japan [11], respectively."
"(Line 61) Skipping breakfast is common in university students [27]. The prevalence of skipping breakfast in university students was 32%, 9-16%, 33%, 46-69%, and 17-31% in the US [28], Spain [29,30], Australia [31], China [32,33], and Japan [34,35], respectively. "
3. On the basis of the conducted analysis, it is possible to estimate only the existence of a relationship between the frequency of alcohol consumption and the frequency of skipping breakfast. Unfortunately, the cause-and-effect relationship cannot be assessed. Because maybe it is the more frequent consumption of alcohol that causes skipping breakfasts? Which may be logical, because with symptoms of excessive alcoholism, reluctance to eat is typical.
To control the reverse causality, the present study enrolled the students who drank ≤3 days/week. For example, a male student drank alcohol 3 days/week and had breakfast every day at the baseline visit. Two years after the baseline visit, he drank alcohol 7 days/week and was diagnosed as frequent alcohol drinking. Because of his excessive alcoholism, he skipped breakfast usually at the diagnosis of frequent alcohol drinking.
In this present study, he was categorized into the “eating every day” group in Table 1. In the statistical models used in the present analyses, information before the incidence of frequent alcohol drinking (having breakfast every day at the baseline visit) was included, but information after the incidence of frequent alcohol drinking (skipping breakfast usually 2 years after the baseline visit) was never considered. Thus, the results of the present study strongly suggest that skipping breakfast affected frequent alcohol drinking and do not suggest that frequent alcohol drinking affect breakfast skipping.
4. I do not understand the reason for excluding students who consume alcohol ≥ 4 times a week?
The outcome of interest in the present study was the incidence of frequent drinking of alcohol (≥4 days per week). Thus, we excluded the frequent drinkers of alcohol at the baseline visit, and the student without the outcome at the baseline visit.
5. It is not clear to me why questions were asked about sleep duration, stress levels or smoking? Have relationships between these factors been studied? If so, it should be noted in Introduction. If not, then perhaps this information should be omitted.
Because these covariates were potential confounding factors, we included them as covariates in multivariable-adjusted Cox proportional hazards models. We clearly described their potential confounding in 2.2 Measurements as follows.
"(Line 95) The baseline variables at the health checkup on admission included age, body mass index (BMI = body weight [kg]/height2 [m2]), and general questionnaires regarding breakfast frequency, frequency of drinking alcohol, and potential confounders, includ-ing smoking status [40,41], stress frequency [42,43], weekday sleep duration [44,45], and living arrangements [27,46].)
6. Line 123 (and in Table 1,2,3): “skipping ≥ often” – please explain?
We are very sorry for immature description of the categorization of breakfast frequency. We explain its categorization in detail as follows:
"(Line 131) Owing to the small number of students skipping breakfast usually (male and fe-male students, n = 201 [1.2%] and 76 [0.9%], respectively), we combined those skipping breakfast usually and those skipping often into a single category. Thus, breakfast frequency was recategorized into three groups: eating every day, skipping occasionally, and skipping often/usually."
7. There are different fonts size in tables 1, 2 and 3.
We're sorry for our careless mistakes. We reorganized tables.
8. Due to the very small numbers, it may not be worth analyzing the data into age categories (e.g. in the case of female skipping breakfast ≥often only 3!). Maybe it is worth doing an analysis for the age groups, e.g. 18-19, 20-22?
According to the reviewer's suggestion, we categorized age into 3 categories: 18, 19, and 20-22 years and included them in the multivariable-adjusted hazard ratios in male and female students. Adjusted hazard ratio of eating every day, skipping occasionally, and skipping ≥often were 1.00 (reference), 1.02 (95% confidence interval, 0.84, 1.25), and 1.47 (1.16, 1.86), respectively, in male students and 1.00 (reference), 1.60 (1.03, 2.49), and 3.12 (1.87, 5.21), respectively, in female students. The results were almost same as those shown in Table 3. We did not add these models to Table 3 because the age categorization did not affect the association between breakfast skipping and the incidence of frequent drinking of alcohol. If the reviewer and the editors will need the table of the results of multivariable-adjusted Cox proportional hazards models for the publication, we are ready to add the results of this model to Table 3 as "model 2", labeling the adjusted model in the table 3 in this revised manuscript as "model 1."
9. The results of the statistical analysis should be shown in the tables!
The statistical methods described in 2.3 statistical analysis were described in table 1, 2, and 3, and Figure 1. I have no idea what we should add to this revised manuscript. We would like the reviewer to clarify the results which we should add to this revised manuscript.
10. For the reader's convenience, it would be better to distinguish the Conclusions as a separate paragraph.
Based on the comment by the reviewer, we add the subtitle, "(Line 272) 6. Conclusion."
Reviewer 2 Report
An important article by topic, but with a largely subjective assessment of the studied features and no clinical trials to explain the relationship. Groups of very different sizes were also compared. The main doubts, however, remain methodically unchanged at the present time, and the authors are aware of this fact.
Author Response
1. An important article by topic, but with a largely subjective assessment of the studied features and no clinical trials to explain the relationship. Groups of very different sizes were also compared. The main doubts, however, remain methodically unchanged at the present time, and the authors are aware of this fact.
We appreciated critical comments by reviewer 2.
One of the disadvantages of the present study was subjective answer to the question of breakfast frequency. Thus, we could not assess a cutoff value of the breakfast frequency at the risk of frequent drinking of alcohol. However, the present study clarifies the dose-dependent association between breakfast frequency and the incidence of frequent drinking of alcohol. In our previous retrospective cohort study, including in 5439 female workers at Osaka University, the same question of breakfast frequency showed a dose-dependent association between breakfast frequency and the incidence of proteinuria (Tomi, R. Nutrients 2020;12:3549). The results of our previous study and the present study strongly suggest that this subjective questionnaire was clinically useful to show the dose-dependent association of breakfast frequecy. We have recently modified this questionnare to ask how many days per week students and workers in Osaka university have breakfast. Using this new questionnaire, we will reassess the dose-dependent association between breakfsat frequency and frequent drinking of alcohol in future.
We described this disadvantage in Discussion as follows.
"(Line 256) Although the present study could not assess a cutoff value for the frequency (days/week) of skipping breakfast to predict the incidence of frequent alcohol drinking, it clarified the dose-dependent association between breakfast frequency and the incidence of frequent alcohol drinking. Our previous retrospective cohort study, using the same questionnaire of breakfast frequency, disclosed a similar dose-dependent association between breakfast frequency and the incidence of proteinuria in 5,439 female workers [26]. The results of the present study and our previous study strongly suggest the subjective questionnaire of breakfast frequency have a clinical relevance. The optimal cutoff value of the frequency (day/week) of skipping breakfast should be assessed in future studies."
Because of the observational study nature of the present study, the number of participants in 3 categories of breakfast frequency were very different. To validate the findings of the present study, randomized controlled trials were essential.
We added the following sentence to Discussion.
"(Line 267) Forth, because of the nature of the observational study design of the present study, the association between breakfast frequency and the incidence of frequent alcohol drinking was critically confounded by unmeasured factors. Clinical impact of breakfast frequency should be verified by randomized controlled trials."
Reviewer 3 Report
Thank you that you give me the opportunity to review this manuscript “ Skipping breakfast and incidence of frequent drinking in university students: A retrospective cohort study”.
The authors presented the interesting, actual topic of skipping breakfast and more frequent drinking among young students.
The topic is noteworthy, and worth further research in other countries.
Some of the comments on this manuscript.
- Manuscript title
should add
Skipping breakfast and incidence of frequent drinking in university students in Japan.
because a large cohort study only covers Japanese students, and cultures and customs are different from those in Europe or the United States.
And
frequent drinking of alcohol
because the word “ drinking” does not synonymous with drinking alcohol
drinking - this is fluid intake, and may be related to drinking of the water, juice, milk, etc.
Especially it is also not marked in key words where the word “alcohol “ is missing.
- Abstract
Line 20: Drinking frequency what drinking? And the same Line 23, and 33
- Introduction
Line 39: ….cancer . – what kind of cancer ? all cancers ?
Line 41: “ Recently, binge drinking in university students has been regarded as a serious social problem”. Where in Asia? Africa? Be more precise when writing for an international journal. You cannot write generalities.
Line 42: “ Multiple studies showed that university students with binge drinking were at high risk …”
If you write "multiple studies" please include more than one reference at the end of this sentence.
Line 49-52- There are no references!!
In the introduction, there is no epidemiological data on alcohol consumption among young students in the world in general and in Japan. Is there a problem with drinking alcohol in Japan? The potential reader did not find out anything in the introduction. Maybe it is forbidden to drink alcohol in this country at all?
Line 64- The findings of the present study shed light…..
it is a colloquialism and does not fit in with a medical journal
Line 63, 65 drinking … what?
2.2 Measurements
Line 95-96 “How often did you have breakfast during the past year?”, for which the possible responses were “eating almost every day,” “skipping occasionally,” “skipping often,” and “skipping usually”.
Answers such as “often, usually, occasionally”, usually cause me anxiety, because it is difficult to compare it with another study, if we do not know exactly what the authors had in mind when writing often, e.g. 2 times a week, 4 times a week. The answers to the questions formulated in this way are very subjective and do not have to coincide with what the young students mean. Even more so when it comes to a habit. In the introduction, we did not find out if the Japanese usually eat breakfast at home, maybe, just like Italians in hot summer, they only drink coffee outside the home.
Line 97: Drinking frequency was based on the question, ‘‘How often did you drink the past year?’’ with five possible responses: “I drank occasionally,”
and again the same, especially when it comes to questions about drinking alcohol.
“I drank occasionally,” - what do you mean?
Alcohol abusers will consider this frequency of alcohol consumption to be rare, in contrast to people who do not drink alcohol at all, it will be very common.
Line 107: …..with five possible responses of <5, 5–6, 6–7, 7–8, and ≥8 h.”
If the student sleeps 6 hours, which answer should choose: 5-6 or 6-7? but if he sleeps 8 hours, what choose 7-8 or ≥8 hours?
Line 114: you write “…drinking frequency ≥4 days/week” but in Line 47 …..is reported that drinkers with drinking frequency ≥3 days/week. Please explain it.
4. Discussion
In the discussion, the authors did not refer to the presented results. This applies to smoking, BMI, stress, hours of sleep, or living arrangement. Certainly, people who live with their families eat breakfast more often.
Please correct the manuscript thoroughly.
Author Response
We appreciate a long list of valuable comments by the reviewer. Your comments remarkably helped us raise the quality of our manuscript. Now we wish that our manuscript is satisfactory to be published in Nutrients.
1. Skipping breakfast and incidence of frequent drinking in university students in Japan. because a large cohort study only covers Japanese students, and cultures and customs are different from those in Europe or the United States.
Thank you for your kind advice. We changed the title as “Skipping breakfast and incidence of frequent alcohol drinking in university students in Japan: A retrospective cohort study.”
2. And frequent drinking of alcohol
because the word “drinking” does not synonymous with drinking alcohol
drinking - this is fluid intake, and may be related to drinking of the water, juice, milk, etc.
Especially it is also not marked in key words where the word “alcohol “ is missing.
Thank you for your useful advice. We added “frequency of alcohol drinking” to keywords.
3. Line 20: Drinking frequency what drinking? And the same Line 23, and 33
We appreciate your kind comments. We clarified that drinking alcohol is the main issue of the present manuscript.
“(Line 19) Frequency of alcohol drinking is a potential predictor of binge drinking of alcohol, a serious so-cial problem in university students.”
“(Line 32) University students who skipped breakfast were at a higher risk of frequent alcohol drinking than those who ate breakfast every day.”
Besides these sentences, we modified some sentences to clarify the target of the present study was “alcohol drinking.”
4. Line 39: ….cancer . – what kind of cancer ? all cancers ?
According to the reference 5, we listed the cancers with dose-dependent association with alcohol consumption as follows:
“(Line 36) Excessive alcohol consumption is a global health and social burden because it is as-sociated with suicide [1], death by injury [2], liver cirrhosis [3], ischemic heart disease [4], and cancers, including those of oral cavity and pharynx, esophagus, colorectum, liver, larynx, and female breast [5].”
5. Line 41: “Recently, binge drinking in university students has been regarded as a serious social problem”. Where in Asia? Africa? Be more precise when writing for an international journal. You cannot write generalities.
We appreciate your valuable comment. The binge drinking is very common in university students and regarded as a serious social problem in developed countries, including the US and Japan. Thus, we modified the sentences as follows.
“(Line 39) Binge drinking, a type of excessive alcohol consumption defined as consuming five or more standard drinks per occasion for men and four or more drinks for women [6], is common in university students in many countries: 43-44%, 56%, 76%, 26-28%, 23%, and 53% of university students had a recent history of binge drinking in the US [7], the UK [8], Switzerland [9], China [10], and Japan [11], respectively. Recently, binge drinking in university students has been regarded as a serious social problem in these countries [11,12], because multiple studies showed that university students with binge drinking were at high risk of missed classes and lower grades [13]; injuries and sexual assaults [14]; non-medical use of prescription drugs [15]; memory blackouts [16]; changes in brain function [17]; lingering cognitive deficits [18]; and even death [14].”
6. Line 42: “Multiple studies showed that university students with binge drinking were at high risk …”If you write "multiple studies" please include more than one reference at the end of this sentence.
The Reference in the previous manuscript is a good narrative review, which described the clinical impact of binge drinking in university students, based on many papers. As the reviewer recommended, multiple references were appropriate. We modified the sentence as follows.
“(Line 43) Recently, binge drinking in university students has been regarded as a serious social problem in these countries [11,12], because multiple studies showed that university students with binge drinking were at high risk of missed classes and lower grades [13]; injuries and sexual assaults [14]; non-medical use of prescription drugs [15]; memory blackouts [16]; changes in brain function [17]; lingering cognitive deficits [18]; and even death [14].”
7. Line 49-52- There are no references!!
We're sorry for our immature description. The sentence was based on the reference cited in the sentence just before the sentence. We combined these two sentences as follows.
“(Line 50) A large cross-sectional study, including 10,466 current drinkers aged 18–76 years in Canada, reported that drinkers with ≥3 days/week of alcohol drinking were more vul-nerable to binge drinking than those with <3 days/week of alcohol drinking and, addi-tionally, suggested that the association between frequency of alcohol drinking and binge drinking was stronger in younger drinkers than that in older drinkers [19].”
8. In the introduction, there is no epidemiological data on alcohol consumption among young students in the world in general and in Japan. Is there a problem with drinking alcohol in Japan? The potential reader did not find out anything in the introduction. Maybe it is forbidden to drink alcohol in this country at all?
We thank the reviewer for this valuable comment. We added the prevalence of binge drinking and skipping breakfast as follows.
"(Line 39) Binge drinking, a type of excessive alcohol consumption defined as consuming five or more standard drinks per occasion for men and four or more drinks for women [6], is common in university students in many countries: 43-44%, 56%, 76%, 26-28%, 23%, and 53% of university students had a recent history of binge drinking in the US [7], the UK [8], Switzerland [9], China [10], and Japan [11], respectively."
"(Line 61) Skipping breakfast is common in university students [27]. The prevalence of skipping breakfast in university students was 32%, 9-16%, 33%, 46-69%, and 17-31% in the US [28], Spain [29,30], Australia [31], China [32,33], and Japan [34,35], respectively. "
In Japan adult aged 20 years or older can drink by law. We provide this information in Introduction as follows.
"(Line 69) The present 5-year retrospective cohort study aimed to clarify the association be-tween skipping breakfast and the incidence of frequent alcohol drinking in 26,179 university students at a single national university in Japan, where drinking alcohol under the age of 20 is prohibited by law."
9. Line 64- The findings of the present study shed light…..it is a colloquialism and does not fit in with a medical journal
We changed the sentence as follows.
“(Line 72) The findings of the present study clarify the clinical impact of skipping breakfast on the frequency of alcohol drinking in young adults.”
10. Line 63, 65 drinking … what?
In this revised manuscript, we modified “drinking” to indicate “alcohol drinking,” including the title. The sentences indicated by the reviewer were rewritten as follows.
“(Line 69) The present 5-year retrospective cohort study aimed to clarify the association be-tween skipping breakfast and the incidence of frequent alcohol drinking in 26,179 uni-versity students at a single national university in Japan, where drinking alcohol under the age of 20 is prohibited by law. The findings of the present study clarify the clinical impact of skipping breakfast on the frequency of alcohol drinking in young adults.”
11. Line 95-96 “How often did you have breakfast during the past year?”, for which the possible responses were “eating almost every day,” “skipping occasionally,” “skipping often,” and “skipping usually”.
Answers such as “often, usually, occasionally”, usually cause me anxiety, because it is difficult to compare it with another study, if we do not know exactly what the authors had in mind when writing often, e.g. 2 times a week, 4 times a week. The answers to the questions formulated in this way are very subjective and do not have to coincide with what the young students mean. Even more so when it comes to a habit. In the introduction, we did not find out if the Japanese usually eat breakfast at home, maybe, just like Italians in hot summer, they only drink coffee outside the home.
We appreciate the reviewer’s critical comment on the great disadvantage of our study. The answer to the questionnaire of the breakfast frequency was so subjective that it was difficult to make the cutoff value of breakfast frequency for the incidence of frequent drinking alcohol and to compare the finding of the present study with other studies. However, the results of the present study strongly suggest the dose-dependent association between breakfast frequency and the incidence of frequent drinking of alcohol. In our previous retrospective cohort study, using the same questionnaire of breakfast frequency as that in the present study, showed a similar dose-dependent association between breakfast frequency and the incidence of proteinuria in 5439 female university workers. These findings of our previous study and the present study suggest that the questionnaire used in the present study were clinically relevant, although the answer was subjective.
In Osaka University the questionnaire of breakfast frequency has been changed to ask how many days students and workers have breakfast per a week. We’ll assess the association between the exact number of days per week having breakfast and the incidence of frequent drinking alcohol in future.
We added the following sentences to Discussion.
“(Line 257) Although the present study could not assess a cutoff value for the frequency (days/week) of skipping breakfast to predict the incidence of frequent alcohol drinking, it clarified the dose-dependent association between breakfast frequency and the inci-dence of frequent alcohol drinking. Our previous retrospective cohort study, using the same questionnaire of breakfast frequency, disclosed a similar dose-dependent associa-tion between breakfast frequency and the incidence of proteinuria in 5,439 female workers [25]. The results of the present study and our previous study strongly suggest the subjective questionnaire of breakfast frequency have a clinical relevance. The opti-mal cutoff value of the frequency (day/week) of skipping breakfast should be assessed in future studies.”
Japanese people mainly eat rice or bread at breakfast. We tried hard to search for the paper reporting the prevalence of eating out at breakfast in Japan, but we could not find it. We added the following sentence to Introduction to describe the Japanese general breakfast.
"(Line 57) Japanese people mainly eat rice or bread at breakfast [20]. "
12. Line 97: Drinking frequency was based on the question, ‘‘How often did you drink the past year?’’ with five possible responses: “I drank occasionally,”
and again the same, especially when it comes to questions about drinking alcohol.
“I drank occasionally,” - what do you mean?
Alcohol abusers will consider this frequency of alcohol consumption to be rare, in contrast to people who do not drink alcohol at all, it will be very common.
In contrast to the white and black people, approximately 5% of Asian people cannot drink alcohol genetically, because the activity of the enzyme to metabolize alcohol is genetically very low. Thus, students with genetically low activity of the enzyme did not drink alcohol at all and were categorized into "I did not drink" in the present study.
The answer of "I drank occasionally" was for those who could drink alcohol genetically and drank sometimes at the frequency below the level of "I drank a day/week."
In the present study, drinking frequency of alcohol was divided into two categories of “I drank ≥4 days/week” or not. Thus, we assume that the definition of "I drank occasionally" did not affect the main finding of the present study.
13. Line 107: …..with five possible responses of <5, 5–6, 6–7, 7–8, and ≥8 h.”
If the student sleeps 6 hours, which answer should choose: 5-6 or 6-7? but if he sleeps 8 hours, what choose 7-8 or ≥8 hours?
This questionnaire asked average sleep hours on weekdays. A student with 7:30 of average sleep duration on weekdays probably chose "7–8 h". Another student with 7:59 of average sleep duration on weekdays probably chose "7–8 h." The problem of this questionnaire was which categories the students with 7:00 of average sleep duration on weekdays choose, "7–8 h" or "8–9 h,"? Students with 6:00 and 8:00 of average sleep duration had the same problem. However, we assume the proportion of those with average sleep duration of just 7:00, neither 6:59 nor 7:01, were small, suggesting this problem in students with 6:00, 7:00, and 8:00 of average sleep duration on weekdays affect the main findings of the present study.
14. Line 114: you write “…drinking frequency ≥4 days/week” but in Line 47 …..is reported that drinkers with drinking frequency ≥3 days/week. Please explain it.
The previous Canadian study showed that adults with drinking frequency of ≥3 days per week were vulnerable to binge drinking (reference 13). Because the category of frequency of drinking alcohol in the present study were different from those of this Canadian study, we regarded the drinking frequency of ≥4 days per week, included in ≥ 3 days per week, as a risk factor of binge drinking.
We added the following sentence to 2.2. Measurements.
"(Line 123) We set the outcome at ≥4 days/week of alcohol drinking because the previous Canadian cohort study showed that adults with ≥3 day/week of alcohol drinking were vulnerable to binge drinking [19]. "
15. In the discussion, the authors did not refer to the presented results. This applies to smoking, BMI, stress, hours of sleep, or living arrangement. Certainly, people who live with their families eat breakfast more often.
We discussed the findings of the presents study in Discussion. To stress the results of the present study, we modified the sentence as follows.
"(Line 222) The present retrospective cohort study, which included 17,380 male and 8,799 female university students, verified that skipping breakfast predicted the incidence of frequent alcohol drinking in both male and female students (Table 3 and Figure 2), suggesting that university students who skip breakfast might be vulnerable to the detrimental ef-fects of excessive alcohol consumption."
Smoking, BMI, stress, sleep duration, and living arrangement were covariates, not the main exposure of the present study. Thus, we did not discuss their associations with the incidence of frequent alcohol drinking.
16. Please correct the manuscript thoroughly.
According to many valuable comments by reviewers, we revised our manuscript. We greatly appreciate your and other reviewers' kind cooperation to our paper.